# Implementation of a New Protocol for Direct Identification from Urine in the Routine Microbiological Diagnosis

**DOI:** 10.3390/antibiotics11050582

**Published:** 2022-04-26

**Authors:** Yuliya Zboromyrska, Verónica Rico, Cristina Pitart, Mariana José Fernández-Pittol, Álex Soriano, Jordi Bosch

**Affiliations:** 1Microbiology Department, Hospital Clinic, 08036 Barcelona, Spain; yzboromy@clinic.cat (Y.Z.); cpitart@clinic.cat (C.P.); mjfernandez@clinic.cat (M.J.F.-P.); 2Infectious Diseases Department, Hospital Clinic, 08036 Barcelona, Spain; rico@clinic.cat (V.R.); asoriano@clinic.cat (Á.S.); 3Barcelona Institute for Global Health (ISGlobal), Hospital Clinic, University of Barcelona, 08036 Barcelona, Spain

**Keywords:** urine, MALDI-TOF, flow cytometry, urinary tract infection, ESBL, carbapenemases

## Abstract

Background: The direct identification of uropathogens from urine samples, in combination with the rapid detection of resistance, would allow early adjustment of empirical antimicrobial treatment. Methods: Two hundred and ninety-eight urine samples processed between 1 June and 31 December 2020, selected with flow cytometry, with direct identification by MALDI-TOF mass spectrometry, and rapid detection of extended-spectrum beta-lactamase (ESBL) and carbapenemases-producing strains by lateral flow were analyzed. Results: The positive predictive value of the direct identification of the 86 samples that met the flow cytometry criterion (>5000 bacteria/µL) was 96.4%. Reliable direct identification was obtained in 14 of the 27 (51.8%) urinary source bacteraemias. There was 100% agreement between the lateral flow and antibiogram in the detection of ESBL and carbapenemases. Conclusion: the protocol for the direct identification and rapid detection of ESBL and carbapenemases-producing strains from urine samples is a reliable and useful tool.

## 1. Introduction

Urinary tract infection (UTI) is one of the most common infections in both the community and hospital settings. The challenge of administering correct antimicrobial treatment is important, taking into account the increase in resistance among the pathogens that most frequently cause UTI, especially the production of extended-spectrum beta-lactamases (ESBLs) and carbapenemases in *Enterobacterales* [1,2].

Matrix-assisted laser desorption/ionization time-of-flight mass spectrometry (MALDI-TOF MS) is currently the main tool for the routine identification of microorganisms in the clinical microbiology laboratory. Its potential for direct identification from urine samples has been demonstrated in numerous studies [3,4,5]. This technique was previously evaluated by our group, and a combined protocol of flow cytometry and MALDI-TOF MS was proposed for direct identification in urine samples. The usefulness of the proposed protocol was later evaluated in a multicenter study [6,7]. Thereafter, the protocol was launched as part of routine diagnosis in our hospital. The aim of this study was to describe our experience and the results obtained during the first seven months of implementation of the direct identification of urine samples, together with the direct application of rapid detection techniques to identify ESBL and carbapenemases-producing strains.

## 2. Materials and Methods

### 2.1. Samples

Two hundred and ninety-eight urine samples, collected without preservative, were analyzed between 1 June and 31 December 2020. In this first phase of the introduction of the new protocol, only samples from the Emergency Department of the hospital, accompanied by blood cultures as an indicator of the suspected bacteremia, were analyzed. It was necessary for these samples to have a sample volume of ≥10 mL and a bacterial count according to the flow cytometer of ≥5000 bacteria/µL and to have arrived at the laboratory between 0:00 a.m. and 3:00 p.m. for processing within the routine working hours of the laboratory (from 8:00 a.m. to 3:00 p.m.).

### 2.2. Flow Cytometry

The samples were processed with the UF-1000 flow cytometer (Sysmex, Kobe, Japan), according to the manufacturer’s instructions.

### 2.3. Culture and Antibiotic Susceptibility Study

The samples were plated quantitatively on cystine lactose electrolyte deficient agar (Becton Dickinson GmbH, Heidelberg, Germany) and incubated for 48 h at 37 °C in an aerobic atmosphere. The microorganisms isolated in culture were identified by MALDI-TOF MS (Bruker Daltonik, GmbH, Bremen, Germany). The culture was considered as positive if a significant growth (≥10^4^ colony forming units (CFU)/mL) of one or two microorganisms were detected. Cultures with three or more microorganisms were reported as contaminated. The antimicrobial susceptibility study was performed using the disk diffusion method and the results were interpreted according to the latest EUCAST guideline (http://www.eucast.org) (v.10.0 accessed on 1 January 2020).

### 2.4. Direct Identification by MALDI-TOF MS

The following protocol was used to achieve direct identification by MALDI-TOF MS. First, 10 mL of urine was centrifuged at 430× *g* for 5 min; then, the supernatant was centrifuged at 15,600× *g* for 2 min; the pellet obtained was washed twice with sterile water and used for identification by MALDI-TOF MS. Each sample was analysed in duplicate. For this, two spots of MSP 96 target polished steel plates (Bruker Daltonik GmBH) were covered with 1 μL of the pellet obtained and air-dried. Then, both spots were covered with 1 μL of matrix solution (cyano-4-hydroxy-cinnamic acid) in 50% acetonitrile with 2.5% trifluoroacetic acid (Bruker Daltonik GmBH). Spectra acquisition was performed in MALDI Microflex LT (Bruker DaltoniK GmbH), using flexControl v.3.0 software. The final ID was achieved using Biotyper v.3.0 software. According to the manufacturer, a score of <1.7 indicates no reliable identification (NRI), a score between 1.7 and 2.0 indicates genus identification, and a score of ≥2.0 indicates species identification. Apart from a score value, MALDI-TOF MS software provides a list of ten microorganisms with the most similar spectra. We considered the species-level ID valid if the same species with a score of ≥1.7 was obtained for the first microorganism from the list of the two dropped spots or for the first two microorganisms from the list of the same spot, if one of the spots was NRI or No Peaks (NP). The highest score obtained from two spots was recorded to calculate the average score value [6].

### 2.5. Direct Detection of Resistance (ESBL and Carbapenemases)

In the case of obtaining reliable identification of *Escherichia coli*, *Klebsiella pneumoniae* complex (including *K. pneumoniae*, *K. quasipneumoniae*, and *K. variicola* species), and *Enterobacter cloacae* complex (including *E. asburiae*, *E. cloacae*, *E. hormaechei*, *E. kobei* and *E. ludwigii*), the NG-Test CTX-M MULTI, a rapid immunochromatography technique (lateral flow), which detects the CTX-M-type enzymes belonging to groups 1, 2, 8, 9 and 25, and the NG-Test CARBA 5, which detects the carbapenemases NDM, VIM, IMP, KPC and OXA-48-like (both from NG biotech, Guipry, France), were performed according to the manufacturer’s instructions. The processing time was one minute and the results were read after 15 min.

### 2.6. Communication of Results

In the case of obtaining reliable identification, the result of the identification and the detection of resistance (for the indicated species) were reported to the assigned infectious diseases specialist, who reviewed the patient’s clinical history and the empirical treatment received by the patient. The results of flow cytometry, as well as direct identification by MALDI-TOF MS, were entered and validated in the laboratory information system.

## 3. Results

The results obtained are summarized in Figure 1.

The positive predictive value of the direct identification of the 86 samples that met the cytometry criteria was 96.4% (54/56), with 95% CI = 88.7–98.9%. Fifty-four (47.4%) of one hundred and fourteen culture-positive samples (*n* = 78 from flow cytometry with ≥5000 bacteria/μL and *n* = 36 from flow cytometry with <5000 bacteria/μL) were correctly identified on the same day of arrival at the laboratory. Table 1 shows the species identified, the direct MALDI-TOF MS value and the result of the urine culture. The average direct identification value was 2.198.

Upon analyzing the 23 samples, in which a reliable direct identification was not achieved despite being positive by urine culture, two microorganisms were identified in four samples, one microorganism with a count of <10^5^ CFU/mL was isolated in culture in three samples, and in the other three the leukocyte count was >10,000 cells/µL. In the remaining samples, the possible cause for the failure of direct identification was not found.

Two discordant results (2.3%) were obtained in the culture results of the eighty-six samples processed for direct identification by MALDI-TOF MS. In the first case, *Lactobacillus crispatus* was directly identified, while the *K. pneumoniae* complex and *Candida albicans* were isolated in culture. In addition, another discordant result corresponded to the direct identification of *E. coli*, while the urine culture was reported as contaminated.

Of the 212 samples that did not meet the criteria of a count of ≥5000 bacteria/µL, 36 were found to be culture positive, representing 31.6% of all positive samples. It is noteworthy that 22 of the 36 samples (61.1%) had counts of <10^5^ CFU/mL.

Regarding the detection of resistance, out of 54 samples with reliable and positive identification by culture, ESBL and carbapenemases were tested in 37 with a 100% concordance between the rapid techniques and the conventional antibiogram. No carbapenemase-producing strain was detected and ten ESBL-producing strains were correctly identified, including six *E. coli* and four *K. pneumoniae* complex.

Regarding the 54 cases reliably and positively identified by culture, in 38 cases (70.3%), the patient received adequate empirical therapy and the treatment was not modified, while in 16 cases (29.7%) the treatment was modified, according to the results of direct identification and detection or not of ESBL and/or carbapenemases.

In relation to the empirical treatment in ten cases of the ESBL-producing strains, in four patients ceftriaxone was changed to ertapenem after receiving the results from the Microbiology Department. Two of these four patients were immunosuppressed (a heart transplant patient and a HIV patient) and presented bacteremia from a urinary focus, with the same strain being isolated in the blood culture. The remaining six patients were receiving adequate empirical coverage: four with ertapenem due to a history of ESBL-producing strains and two with meropenem. Regarding the cases in whom the rapid techniques were not performed because they were not the indicated species, no ESBL or carbapenemase-producing strain was identified by conventional antibiogram.

Of the 298 blood cultures collected on the same day as the urine sample, 27 were positive for the same species isolated in the urine culture: 14 (51.8%) showed concordant results between blood culture, urine culture and direct identification; in 8 reliable direct identification was not obtained, in 4 direct identification was not performed because the bacterial count by cytometry was < 5000/µL, and, finally, one case corresponded to one of the two discordant results mentioned above.

## 4. Discussion

The need for rapid diagnostic tools in microbiology is increasing. Relevant microbiological results obtained by a direct identification protocol from urine would allow early adjustment of empirical antimicrobial treatment. However, the sample preparation process for MALDI-TOF MS is laborious, and the patients in whom this technique is to be applied must be carefully selected. In this study, the protocol was carried out in samples from patients with probable bacteremia from a urinary focus, treated in the Emergency Department of our hospital. An analysis of the results obtained seven months after the incorporation of the protocol in the routine laboratory procedures showed a lower rate of reliable identification among the samples that met the criteria of cytometry, and were positive by culture, compared to the previous studies of our group (69.2% vs. 86.4% and 80.3%) [6,7]. The high percentage of unidentified samples (29.5%) may be due to the fact that the test was carried out by different technicians, with variable experience in the processing of samples for direct identification. Nonetheless, almost half of the samples positive by culture were correctly identified the same day of their arrival at the laboratory, including 14 cases of bacteremia from a urinary focus.

The success of direct identification from a biological sample primarily depends on the bacterial count. For this reason, flow cytometry, which provides cells count, shows an important advantage as a screening method prior to the direct identification. According to our data, and in accordance with the results obtained in two previous studies, the MALDI-TOF MS score in the identified samples was quite high (2.198). Therefore, lowering the cut-off point of the MALDI-TOF MS value in direct samples, as suggested in other studies, is not necessary [8,9,10]. Many authors have proposed different strategies to improve the rates of reliable direct identification [11,12,13]. In the present study, the implementation of the normal differential centrifugation protocol allowed the reliable identification of most of the selected samples, within a reasonable period of time (30 min of sample preparation). Any additional step would involve making the technique even more laborious and lengthy.

Only two discordant results were obtained during the studied period. One corresponded to the identification of *E. coli* in a sample later reported as contaminated, according to culture results. The reliable detection of one of the predominant microorganisms in a bacterial mixture has been previously described [5]. The other case corresponded to the direct identification of *Lactobacillus crispatus*, a microorganism that is part of the female genital microbiota. The urine culture was positive for *K. pneumoniae* complex and *C. albicans*. *Lactobacillus* would probably have been found in much higher counts in direct urine, although its growth may have been masked by the other two microorganisms. In a multicenter study, similar results were obtained in some samples, in which microorganisms from the normal or altered microbiota were identified, such as *Gardnerella vaginalis*, *Bifidobacterium* spp. or *Lactobacillus* spp. [7]. Although these results are discordant, the direct detection of these microorganisms would help to assess urine culture findings, which reflect probable contamination of the sample.

The rapid detection of ESBL- and carbapenemase-producing strains provides essential clinical information. In the present study, carbapenemase-producing strains were not detected in urine samples from the Emergency Department, since most of these infections were probably of community origin, where carbapenemase-producing strains are less frequent. Regarding ESBLs, those of the CTX-M family have practically displaced other families, including TEM and SHV, with the variants of the CTX-M groups 1 and 9 being especially frequent [14].

According to the present results, the antibiotic treatment was modified in 30% of the cases with reliable identification. Of the ten cases with positive ESBL detection, the empirical treatment was changed in four. The selection of species for the rapid test, based on local epidemiology, allows for the rational use of the available resources.

In summary, the implementation of a direct identification protocol from urine samples allowed rapid diagnosis in almost 50% of the samples that were subsequently positive by culture, and in more than half of the samples from patients with bacteremia of confirmed urinary focus. The rapid detection of ESBL- and carbapenemase-producing strains allows early review and adjustment of empirically prescribed treatments.

## Figures and Tables

**Figure 1 antibiotics-11-00582-f001:**
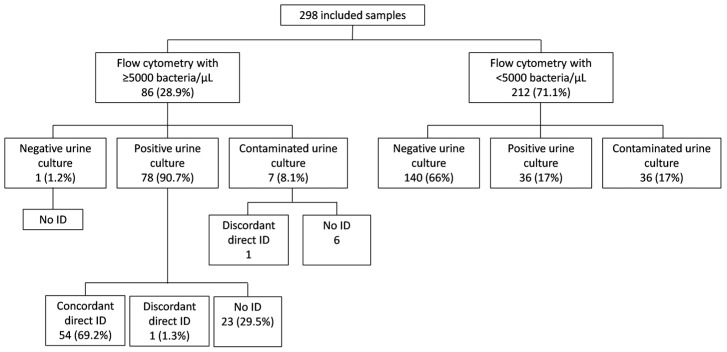
Results of the 298 samples analyzed. ID, identification.

**Table 1 antibiotics-11-00582-t001:** Summary of 54 positive culture samples with reliable direct identification.

Direct ID Using MALDI-TOF (Number of Samples)	Average or Maximum Value of Direct ID (Minimum–Maximum)	Culture Result (Number of Samples)
*Escherichia coli* (30)	2.214 (1.861–2.498)	*E. coli* (27), *E. coli* + *E. faecalis* (1), *E. coli* + *K. pneumoniae* complex (1) and *E. coli* + *P. aeruginosa* (1)
*Klebsiella pneumoniae* complex (10)	2.189 (1.898–2.430)	*K. pneumoniae* complex (8), *K. pneumoniae* complex + *P. aeruginosa* (1) and *K. pneumoniae* complex + *E. faecium* (1)
*Pseudomonas aeruginosa* (3)	2.018 (1.770–2.145)	*P. aeruginosa* (2) and *P. aeruginosa* + *K. pneumoniae* complex (1)
*Klebsiella aerogenes* (2)	2.367 and 2.394	*K. aerogenes* (2)
*Klebsiella oxytoca* (2)	2.291 and 2.377	*K. oxytoca* (2)
*Proteus mirabilis* (2)	1.857 and 2.232	*P. mirabilis* (2)
*Morganella morganii* (1)	2.305	*M. morganii* (1)
*Enterobacter cloacae* complex (1)	2.221	*E. cloacae* complex (1)
*Enterococcus faecalis* (1)	2.334	*E. faecalis* (1)
*Enterococcus faecium* (1)	2.089	*E. faecium* + *E. faecalis* (1)
*Aerococcus urinae* (1)	1.883	*A. urinae* (1)

ID, identification.

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
