# Peer review of "Implementation of a New Protocol for Direct Identification from Urine in the Routine Microbiological Diagnosis"

_antibiotics, 2022, doi:10.3390/antibiotics11050582_

Round 1

Reviewer 1 Report

Overall, the authors reported an interesting study for a protocol to identify uropathogens in the microbiological disgnosis routine. I think this tool described here is useful for the community to detect ESBL and carbapenemase. 

Maybe the sentence in the abstract can be modified as follows.
"METHODS: 298 urine samples processed, selected with flow cytometry, with direct identification by MALDI-ToF mass spectrometry, and rapid detection of extended-spectrum beta-lactamase (ESBL) and carbapenemases producing strains by lateral flow were analyzed"

Author Response

Reviewer 1

The complete manuscript has been corrected by an experienced English-speaking writer.

Maybe the sentence in the abstract can be modified as follows.

"METHODS: 298 urine samples processed, selected with flow cytometry, with direct identification by MALDI-ToF mass spectrometry, and rapid detection of extended-spectrum beta-lactamase (ESBL) and carbapenemases producing strains by lateral flow were analyzed"

Thank you for your suggestion. We have checked recommendation for scientific writing and It seems that numbers beginning a sentence must be spelled out.

Reviewer 2 Report

This manuscript by Yuliya Zboromyrska et al. describes the results of the implementation of a direct identification protocol from urine in the microbiological diagnostic routine of the Barcelona Hospital Clinic.

The manuscript deserves a thorough review.

Global: 

  • Prefer the passive voice.
  • Names of bacteria should be italicized.
  • All numbers below 12 should be written in letters.

Methods: 

  • Are the samples collected on a special medium (such as borate acids).
  • Paragraph 2.4: consider the subheading (lines 69 and 79)
  • The sample preparation protocol should be described, at least briefly, although the reference is helpful, it would make the manuscript easier to read.

Results:

  • For non-users of Bruker MALDI TOF, please give details for interpretation of average or maximum values obtained for IDs.
  • For all performances (sPPV, NPV, concordance,...) indicate the 95%CI.
  • It is necessary to justify the number of subjects/subjects to be included to obtain robust performances. If not, the results could not be considered as robust.
  • A table summarizing concordant or discordant results would be interesting, please consider it.

Discussion: 

  • Because ESBL/Carb were community-acquired, they could be identified in ED patients. The rationale can be considered false (lines 170-176).

Author Response

Reviewer 2

  1. Global: 

The complete manuscript has been corrected by an experienced English-speaking writer.

  1. Prefer the passive voice.

Done

  1. Names of bacteria should be italicized

Done

  1. All numbers below 12 should be written in letters.

All numbers below 12 from Results section have been written in letters as suggested.

  1. Methods: 
  2. Are the samples collected on a special medium (such as borate acids).

You are right. This point has been clarified in the text (2.1 Samples):

Two hundred ninety-eight urine samples, collected without preservative, were analysed between June 1 and December 31, 2020.  

  1. Paragraph 2.4: consider the subheading (lines 69 and 79)

Thank you for pointing this out. We have incorporated this idea and created additional paragraphs: 2.5 and 2.5

  1. The sample preparation protocol should be described, at least briefly, although the reference is helpful, it would make the manuscript easier to read.
    • We agree. We have added this information to the Method section, 2.4 Direct identification

III. Results

  1. For non-users of Bruker MALDI TOF, please give details for interpretation of average or maximum values obtained for IDs.

This point has been clarified in the paragraph 2.4

  1. For all performances (sPPV, NPV, concordance,...) indicate the 95%CI.

Done: The positive predictive value of direct identification of the 86 samples that met the cytometry criteria was 96.4% (54/56), with 95% CI = 88.7-98.9%.

MedCalc® was used as statistical software.

  1. It is necessary to justify the number of subjects/subjects to be included to obtain robust performances. If not, the results could not be considered as robust.

The estimation of the number of samples to be included was done in our previous studies (references 6 and 7), when the protocol was proposed and then validated. The aim of this communication is to describe our experience and the results obtained during the first seven months of implementation of the protocol in routine procedures, including the impact on patient’s management. I am afraid that there is nothing we can do at this point.

  1. A table summarizing concordant or discordant results would be interesting, please consider it.

Only two discordant results were obtained between direct identification and in culture, and no discordant results were obtained for ESBL detection. These two discordant results have been explained in the text.

  1. Discussion: 

Because ESBL/Carb were community-acquired, they could be identified in ED patients. The rationale can be considered false (lines 170-176).

We are sorry for the confusion. We did not explain this clearly. This point has been clarified in the revised text.

Reviewer 3 Report

Zboromyrska and colleagues reported the identification of uropathogens from direct urine samples in combination with the rapid detection of resistance. The work will help accelerate the early adjustment of empirical antimicrobial treatment. I have the following comments for the authors.

Line 15- Please be consistent with the abbreviation for MALDI-TOF mass spectrometry either MALDI-ToF MS or MALDI-TOF MS
Lines 61-63- Rephrase for clarity.
Line 68- Describe the protocol briefly. ......criteria, have been previously described [8]. In brief, ..........
Line 69 - Separate this heading by adding numbers or making the text bold.
Line 70-74- Italicize the genus and species names.
Line 74 - Please change "which detects groups 1, 2, 8, 9 and 25 of CTX-M" to  "which detects CTX-M-type enzymes belonging to group 1, 2, 8, 9 and 25"
Line 79 - same comment as line 69.
Line 84 - Expand LIS
Line 90- Please include the following pertaining to the 114 culture positive samples. Fifty-four (47.4%) of 114 culture-positive samples (n=78 from flow cytometry with ≥5000 bacteria/μL and n=36 from flow cytometry with <5000 bacteria/μL) were correctly identified on the same day of arrival at the laboratory.
Lines 96-100- Please rephrase. Split the sentences in small sentences.
Line 101-- Please make it clear that the 86 is from Flow cytometry with ≥5000 bacteria/μL
Lines 153-155--Please include lines pertaining to this in the methods and results section.
Discussion -- Need to include a sentence on how the samples selected using flow cytometry helped in the direct identification.

Author Response

Reviewer 3

The complete manuscript has been corrected by an experienced English-speaking writer.

Line 15- Please be consistent with the abbreviation for MALDI-TOF mass spectrometry either MALDI-ToF MS or MALDI-TOF MS

Done

Lines 61-63- Rephrase for clarity.

As you suggested, we have clarified this point in the text.

Line 68- Describe the protocol briefly. ......criteria, have been previously described [8]. In brief,...

We agree with your comment. The omission of the protocol has been addressed.

Line 69 - Separate this heading by adding numbers or making the text bold.

We have incorporated this idea and created additional paragraphs: 2.5 and 2.5

Line 70-74- Italicize the genus and species names.

Thanks for pointing out this mistake. Done.

Line 74 - Please change "which detects groups 1, 2, 8, 9 and 25 of CTX-M" to  "which detects CTX-M-type enzymes belonging to group 1, 2, 8, 9 and 25"

We have revised the text accordingly.

Line 79 - same comment as line 69.

Done

Line 84 - Expand LIS

Expanded.

Line 90- Please include the following pertaining to the 114 culture positive samples. Fifty-four (47.4%) of 114 culture-positive samples (n=78 from flow cytometry with ≥5000 bacteria/μL and n=36 from flow cytometry with <5000 bacteria/μL) were correctly identified on the same day of arrival at the laboratory.

As you suggested, we added this information. 

Lines 96-100- Please rephrase. Split the sentences in small sentences.

The text has been changed to improve readability.

Line 101-- Please make it clear that the 86 is from Flow cytometry with ≥5000 bacteria/μL

This point was clarified in the text.

Lines 153-155--Please include lines pertaining to this in the methods and results section.

The protocol was described in the section 2.4

Discussion -- Need to include a sentence on how the samples selected using flow cytometry helped in the direct identification.

We have added the sentence: “The success of direct identification from biological sample primarily depends on the bacterial count. For this reason flow cytometry, which provide cells count, shows an important advantage as a screening method prior to the direct identification.”

Best regards,

Jordi Bosch

Microbiology Department

Hospital Clínic of Barcelona (Spain)

Barcelona, April 20, 2022

Reviewer 4 Report

Manuscript ID: antibiotics-1633016

Type of manuscript: Communication

Title: Implementation of a new protocol for direct identification from urine

in the microbiological diagnosis routine.

Authors: Yuliya Zboromyrska, Verónica Rico, Cristina Pitart, Mariana José

Fernández-Pittol, Álex Soriano, Jordi Bosch

In this research communication authors describe the results obtained on 86 urine samples (>= 5000 bacteria/µl as determined by flow cytometry) using a direct identification by Maldi-TOF MS (protocol was described by the authors in reference 6 and adopted in a multicenter study as described in ref. 7), together with lateral flow for the rapid detection of resistance of ESBL and carbapenemase producing strains. Samples were also plated and microorganisms isolated in culture were identified by MALDI-TOF MS. Culture result and direct MALDI-TOF MS identification were compared.

Remarks

41: workflow

60-61: add type of mass spectrometer and software (version) used for identification.

67-68: reference 8 is same as reference 6 in reference list!

General remark on Reference section: Check references as well as the layout (omit duplicated numerical labeling).

67-68: ‘direct identification’ add ‘by Maldi-TOF MS’

92-93: ‘direct identification value’ and identification values presented in Table 1: values are quite meaningless as such and therefore (as a suggestion) explain briefly the MS approach as well as the ID scoring  in M&M section following 67-68.

Reorganize text 67-85 (as a suggestion):  2.4 Direct identification by Maldi-TOF MS/2.5 Direct resistance detection/ 2.6 Communication of results

Author Response

I think Reviewer 4 is the same as Reviewer 3.

The complete manuscript has been corrected by an experienced English-speaking writer.

41: workflow

Thank you. Removed.

60-61: add type of mass spectrometer and software (version) used for identification.

This point has been clarified in the paragraph 2.4

67-68: reference 8 is same as reference 6 in reference list!

General remark on Reference section: Check references as well as the layout (omit duplicated numerical labeling).

Thanks for pointing out this mistake. Corrected.

67-68: ‘direct identification’ add ‘by Maldi-TOF MS’

Done

92-93: ‘direct identification value’ and identification values presented in Table 1: values are quite meaningless as such and therefore (as a suggestion) explain briefly the MS approach as well as the ID scoring  in M&M section following 67-68.

Thank you for this suggestion. We have included the information about MALDI-ToF score in the paragraph 2.4. However, we believe that the data shown in the Table 1 (microorganisms isolated in culture, including samples with two different species and the concordance with direct ID) will make our results clearer and more useful for microbiologists and clinicians.

Reorganize text 67-85 (as a suggestion):  2.4 Direct identification by Maldi-TOF MS/2.5 Direct resistance detection/ 2.6 Communication of results

We agree. We have incorporated this idea and created additional paragraphs: 2.5 and 2.6

Round 2

Reviewer 2 Report

The manuscript has been revised according to my previous comments.